# Characterization of the South Atlantic Anomaly

Khairul Afifi Nasuddin[1], Mardina Abdullah[1,2], Nurul Shazana Abdul Hamid[3]

[1]Centre of Advanced Electronic and Communication Engineering, Universiti Kebangsaan Malaysia, Bangi, 43600, Malaysia
[2]Space Science Center (ANGKASA), Institute of Climate Change, Universiti Kebangsaan Malaysia, Bangi, 43600, Malaysia
[3]School of Applied Physics, Faculty of Science and Technology, Universiti Kebangsaan Malaysia, Bangi, 43600, Malaysia

*Correspondence to*: Nurul Shazana Abdul Hamid (zana@ukm.edu.my)

**Abstract.** This research intends to characterize the South Atlantic Anomaly (SAA) by applying power spectrum analysis approach. The motivation to study the SAA region is due to its nature. A comparison was made between the stations
in the SAA region and outside the SAA region during the geomagnetic storm occurrence (active period) and normal period where no geomagnetic storm occurred. The horizontal component of the Earth magnetic field data for the occurrence of the active period was taken on 11 March 2011 while for normal period on 3 February 2011. The used data sample rate is 1-minute. The outcomes of the research revealed that the SAA region had a tendency to be persistent during both period. It can be said, it experiences this characteristic because of the Earth's magnetic field strength. Through the research, it is found that as the
Earth magnetic field increases, it is likely to show an antipersistent value. This is found in the high latitude region. The lower the Earth magnetic field, the more it shows the persistent value as in the middle latitude region. In the region where the Earth magnetic field is very low like the SAA region it shows a tendency to be persistent.

## 1 Introduction

Electromagnetic radiation and charged particles from the Sun constantly reach the Earth (Domingos et al., 2017). Protons and electrons from the aurora, the high-speed solar wind, the radiation belts, or large solar coronal mass ejections penetrate into the Earth's atmosphere in different regions of the terrestrial magnetosphere (Sinnhuber et al., 2016). On the other hand, the Earth is surrounded by an almost spherical magnetic field, the magnetosphere, and it is a natural shielding of the Earth's surface to solar and galactic cosmic-ray particles of up to several GeV in energy (Ugusto et al., 2016).

The Earth's magnetic field configuration determines the trapping and distribution of energetic ionized particles (Badhwar, 1997). By far the most dominant of these fields is of core origin, accounting for over 97 per cent of the field observed at the Earth's surface and ranging in intensity from about 30 000 nT at the equator to about 50 000 nT at the poles (Sabaka et al., 2002). The interaction of the solar wind with the magnetic field and atmosphere of the Earth causes, among other effects, disturbances in the ionosphere (Andalsvik and Jacobsen, 2014). With the existence of the Earth's magnetic field, it protects
the world from danger such as geomagnetic storm. But, there still exist a region where the Earth's magnetic field is the weakest;

known as the South Atlantic Anomaly (SAA). The region arises due to the offset of the Earth's dipole of about 436 km from the Earth's center towards direction of the southeast Asia (Asikainen and Mursula, 2008).

The SAA describes a low intensity magnetic field area which spans from east of Africa over the Atlantic Ocean to South America (Koch and Kuvshinov, 2015). Its extent area at the Earth's surface is continuously growing since the intensity instrumental measurements are available covering part of the Southern Hemisphere and centered in South America (Pavón-Carrasco and De Santis, 2016). This region of weak magnetic field has expanded over time and also moved west-ward (Cnossen and Matzka, 2016). The existence of the SAA is linked closely with the geomagnetic field distribution (Heynderickx, 1996). From the high atmosphere and close outer space, SAA is seen as a sort of geomagnetic hole where electric and neutral particles can flow from the Van Allen belts and from the magnetosphere into the atmosphere below (Santis and Qamili, 2010).

In this region, the inner Van Allen radiation belt its nearest approach to the Earth's surface. This result a number of energetic particles in the SAA region larger compare to other places. The energetic particles captured by the geomagnetic field can reach to lower altitudes forming a high-radiation region (Zou et al., 2015).

The SAA plays a vital role for spacecraft orbiting in that region. Very few measurements have made in this region, in comparison to the other regions of the world (Federico et al., 2010). This research aims to characterize the SAA region based on the geomagnetic data collected from several observatories outside and inside this region.

The research focuses on characterizing the SAA by using power spectrum analysis method. This study was motivated by the nature of the SAA, in which the area over the SAA is described by an intense radiation near the Earth's surface because of the particularly weak local geomagnetic field. It appears as the preferred way-in for high-energy particles in the magnetosphere, alongside the polar regions. As for satellite and spacecraft orbiting through the SAA, it will be beneficial to have an update of the magnetic field strength as it can provide the regional magnetic intensity in that area. Therefore, a research carried out regarding SAA can be used as a reference in the satellite launch and increase knowledge in the field of geomagnetic field (Nasuddin et al., 2015).

**1.1 Review of SAA and Power Spectrum Analysis**

Space radiation is a very important factor affecting both humans and electronic systems (Konradi et al., 1994). The trapped radiation environment, consisting of large amounts of energetic charged particles, can be potentially harmful to human beings and space vehicles immerged in it (Qin et al., 2014). Geomagnetically trapped ionized particles, mainly electrons and protons, are a hazard to modern spaceflight (Fürst et al., 2009). Furthermore, the reduction of solar energy when it went through the atmosphere indicated that there is an atmospheric turbidity (Gopir et al., 2018). Modern society relies heavily on complex electronic systems mounted in spacecraft, which are exposed to extraterrestrial influences (Zavvari et al., 2014). There are a number of damage spacecraft sustains whilst in orbit. These range from space debris, problems with the vacuum of space, various problems associated with the plasma environment, and various problems explicitly associated with the radiation environment (Heirtzler, 2002). One of the cases is with the International Space Station. It needs an additional shielding to deal with this sort of problem. Other examples are the Hubble Space Telescope that stops data collection whilst passing by the

Deleted: Therefore,

Deleted: this

Deleted: This

SAA. One SAA effect on the DORIS carrying satellites is the shift of the onboard oscillator frequency (Capdeville et al., 2016). It is known that LEO (Low Earth Orbit) space vehicles spend a significant part of time in the SAA area (Grigoryan et al., 2008). Spacecraft in this area will receive the biggest radiation dose effect that is correlated by the intense fluxes of charged particles. Furthermore, astronaut's condition is influenced by the increased radiation in this area. Operators who control affected space vehicles need to know how best to minimize the risk of anomalies which in many cases simply means knowing, with a high degree of accuracy, when and where to turn the systems on and off (Ginet et al., 2007).

Several researches has been conducted regarding SAA region. Study on the distribution of energetic particle fluxes near SAA, based on Kriging Interpolation, was conducted by Suparta et al. (2013). Kriging interpolation was used in this research to forecast the dissemination of solar charged particles in the SAA area. Data from electrons with 30 keV energy level (meped0e1) taken from the National Oceanic and Atmospheric Administration (NOAA) 15 satellite, on 8 September 2003 and 28 October in 2003 are employed (Suparta et al., 2013). The approach of Ordinary Kriging (OK) was selected since it is the finest linear unbiased estimator. Even so, this research also revealed a few dissymmetry of the estimate as well as variance figures meant for every model, particularly in the $0^0$ longitude area. To resolve this difficulty, the robust variogram estimator usage was proposed.

Research on the radiation fields specific to the South Atlantic Anomaly was carried out by Panova et al. (1992). The research involves the explanation of the spatial as well as temporal behavior of the radiation field within the "MIR" space station through its passage of the South Atlantic Anomaly (SAA) region. The calculation of the radiation fields on the MIR station was carried out by means of the dosimeter "Lyulin". The measurements were carried out in the large diameter working compartment of the MIR station (Panova et al., 1992). The experiments was carry out to provide knowledge of the radiation fields precisely to the SAA as it bring valuable information for radiation safety on cosmonauts in the course of a long orbital flight.

One of the study on SAA is on the New Archeomagnetic Directional Records From Iron Age Southern Africa (ca. 425–1550 CE) and Implications for the South Atlantic Anomaly done by Hare et al. (2018). The new fine record comfirms the researchers earlier inferences that the SAA is the most current sign of a recurring event known as flux expulsion which has a serious effect on the manifestation of the Earth's magnetic field. Longer-term data from this region are crucial to understanding the current trend (Hare et al., 2018). In the research, a potential correlation of changes recorded in the African data accompanied by archaeomagnetism jerks were identify.

A research on the future of SAA in addition to implications for radiation harm in space was carried out by Heirtzler (2002). In this research, the SAA showed an important part in the radiation harm which happens close to the world's orbits. The significant plus current changes of the geomagnetic field in the South Atlantic was used in order to assess the size of the SAA up to the year 2000. This forecast pointed out that radiation harm towards spacecraft as well as mankind in space will very much rise plus current, covering a considerably greater geographical region than the present day. In general, this journal provides references for those who are interested in learning SAA.

The method chosen to characterize the SAA is called power spectrum analysis. Research on the spectral and fractal analyses of geomagnetic and riometric antarctic observations and a multidimensional activity index was done by Santis, Franceschi & Perrone (1997). Their study analyzed the geomagnetic and riometric data by using spectral and fractal analyses. A multidimensional index was obtained from this single-point data set to show the local and the global conditions of the magnetospheric activeness. Their work is a good reference of the power spectrum analysis.

Other studies include Fractal dynamics of geomagnetic storms done by Zaourar et al. (2013). In this work, the variations of the horizontal element of the Earth's magnetic field were studied. This was done to detect scaling reaction of the temporal changeability in geomagnetic data preserved by the Intermagnet observatories throughout the Solar Cycle 23. In this paper, the fractal spectral properties of the geomagnetic time series data were analyzed by applying spectral wavelet analysis techniques.

Reseach on applying power spectrum analysis on the occurrence of sudden storm commencements (SSCs) for solar cycles 11 to 22 was done by Mendoza et al. (2003). The data for SSCs comprise solar cycles 11 to 22 (1868-1996) (Mendoza et al., 2003). The Maximum Entropy Method was opted to determine the power spectral density (PSD) of the SSCs time series, in addition to analyze their periodicities in the time series that were smoothed through obtaining a 13-month running mean. The current research reveal the existence of a peak for the occurrence of SSCs between 20 to 30 y.

Jian-hui and Yao-quan (1995) apply power spectrum analysis for spherical volumes in order to investigate clustering of the Large Bright Quasar Sample. In order to avoid the effects of galactic absorption and foreground galaxies, most of the survey regions are located at galactic latitudes higher than $40^0$ (Jian-hui and Yao-quan, 1995). The research was conducted to analyze the Large Bright Quasar Sample. In conclusion, the quasars of this sample is not clustered, or very weakly clustered.

Stations used in the study are situated within as well as outside the SAA area. Comparability will be made to describe the SAA through comparison of data from stations situated within the SAA area and station positions in the middle (mid) latitude area as well as high latitude region. This is done during the active period, which is the period of the geomagnetic storm occurrence and normal period where no geomagnetic storm occurred. The active period and normal period were chosen based on Kp Index and Dst Index.

Comprehensively, these make it an intriguing subject for characterizing the SAA. It can provide a better knowledge of the Earth-Space surrounding. This study can be a reference for future experimental measurements (Abu-jafar et al., 2017). Since a number of spacecrafts suffered hazards while orbiting through the SAA, it is hoped that this research can provide additional information about designing a better spacecraft to undergo the damage.

**2 Methodology**

**2.1 Stations**

Stations involved in the research were located inside and outside the SAA region. The list of the stations based on IAGA code, geodetic latitude and geodetic longitude can be seen in Table 1 while Fig. 1 shows their positions.

Table 1: Stations involved in the research.

| SAA Region | | | |
|---|---|---|---|
| Station Name | IAGA CODE | Geodetic Latitude | Geodetic Longitude |
| Tsumeb | TSU | -19.202 | 17.584 |
| Hermanus | HER | -34.425 | 19.225 |
| Huancayo | HUA | -12.038 | 284.682 |
| Ascension Island | ASC | -7.949 | 345.624 |
| Vassouras | VSS | -22.400 | 316.400 |
| Mid Latitude Region (Outside SAA Region) | | | |
| Station Name | IAGA CODE | Geodetic Latitude | Geodetic Longitude |
| Fresno | FRN | 37.091 | 240.279 |
| Fort Churchill | FCC | 58.760 | 265.911 |
| Newport | NEW | 48.271 | 242.880 |
| Sitka | SIT | 57.061 | 224.669 |
| Victoria | VIC | 48.517 | 236.582 |
| High Latitude Region (Outside SAA Region and Outside Mid Latitude Region) | | | |
| Station Name | IAGA CODE | Geodetic Latitude | Geodetic Longitude |
| College | CMO | 64.871 | 212.139 |
| Resolute Bay | RES | 74.690 | 265.105 |
| Baker Lake | BLC | 64.319 | 263.988 |
| Godhavn | GDH | 69.250 | 306.470 |
| Thule | THL | 77.470 | 290.770 |

There are 15 stations involved, whereby five stations were located in the SAA region, five stations located in the mid
latitude region and another five stations were at the high latitude region. The black circles in Fig. 1 show stations in the SAA
region while the magenta circles shows stations in the mid latitude region, and the blue circles represent stations in the high
latitude region.

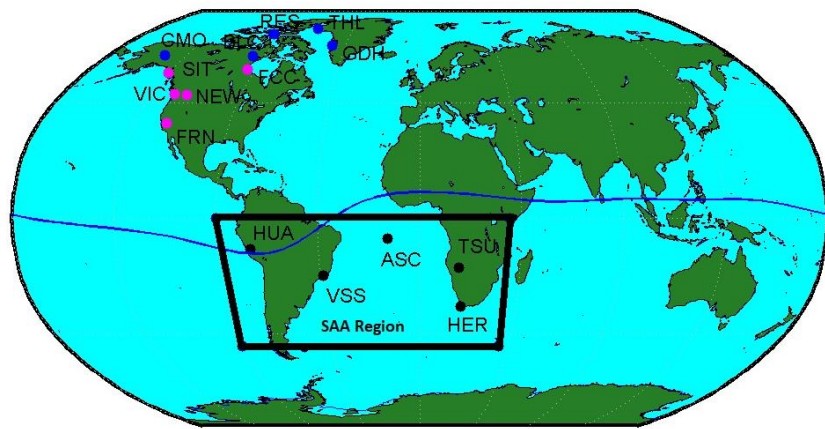

**Figure 1:** SAA is situated at an altitude of 200-800 km over the earth's surface. It extends as of 0 to -50 degrees of latitude along with – 90 to 40 degrees of longitude. The black circle is the station in the SAA region.

**2.2 Power Spectrum Analysis and Hurst Exponent**

The next step is to determine the power spectral density and its scaling with respect to frequency (Hall, 2014). The spectral analysis of a time series allows one to infer something regarding the characteristic time scales of the phenomena which give rise to the observed variations (Santis et al., 1997). A time series can be prescribed either in the time domain, as $y_n$, or in the frequency domain in terms of the discrete Fourier transform, $Y_m$ (Malamud and Turcotte, 1999).

For power-spectral density function, which is represented as $S_m$, whereby it is intended for a discrete time series, represented by $y_n$, n = 1,2,3……..,$N$, can be inscribed as,

$$S_m = \lim_{N \to \infty} \{2 \, |Y_m|^2/N\delta\}, \ \ m = 1,2,3,\dots\dots,\frac{N}{2}, , \tag{1}$$

It can also explain that $\delta$ is the time among consecutive $n$. In another situation, on behalf of a self-affine time series, the power-spectral density, represented as $S_m$, is described to comprise a power-law dependence on frequency

$$S_m \sim f_m^{-\beta}, \ \ m = 1,2,3,\dots\dots,\frac{N}{2}, , \tag{2}$$

It is to be denoted that $f_m = m/N$. It can also be explained that the value of $\beta$ is an estimate of the intensity of persistence in a time series. For $-1 \leq \beta < 1$, the Hurst exponent for a stationary fractional Gaussian noise time series is,

$$H_{PS} = (\beta + 1)/2 \, , \text{for} -1 \leq \beta < 1 , \tag{3}$$

for a nonstationary fractional Brownian motion whereby the $1 < \beta \leq 3$ is represented by,

$$H_{PS} = (\beta - 1)/2 \, , \text{for } 1 < \beta \leq 3 \, , \tag{4}$$

It is important to understand the Hurst exponent, H, value. Time series with 0<H<0.5 are called antipersistent meanwhile time series with 0.5<H<1 are called persistent (Hamid et al., 2009).

If the Hurst exponent is in the range of 0.5<H<1, it can be interpreted as both that a high value in the series will probably be followed by another high value also that the values a long time into the future will serve to be high. As for the Hurst exponent in the range of 0<H<0.5 means a time series with long-term switching between high and low values in adjacent pairs, indicating a single high value will may be succeeded by a low value and the following value will tend to be high, with this trend to change between high and low values, continuing for a long time into the future. While for H = 0.5 implies a

random series. It can also mean data is not correlated, that is no dependence between current and past data.

**Deleted:** interpret

**2.3 Dst index and Kp index for geomagnetic storm period and normal period**

Geomagnetic storm can be known through the Dst index and Kp index. It is to be noted that the red line in the Dst index is the threshold for a geomagnetic storm to occur. As the Dst Index is below -30 nT, it shows a geomagnetic storm occurrence . On 11 March in 2011, the Dst index shows a moderate storm, occurring from 1 to 14 UT, 18 to 19 UT and 21 to

24 UT. During 15 to 17 UT and on 20 UT a weak storm occurred. The Kp Index for the active period showed that on 11 March in 2011 a moderate geomagnetic storm occurred, with Kp Index showing at 6, and a minor geomagnetic storm occurred, with Kp Index showing at 5. Kp Index can be interpreted to show a geomagnetic storm had occurred when it has a value of 5 or larger than 5.

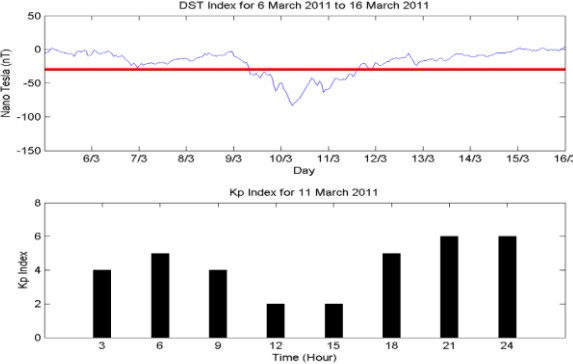

**Figure 2: Dst Index for 6 to 16 March 2011 and Kp Index for active period on 11 March 2011.**

For the normal period, on 3 February in 2011, the Dst Index indicate no geomagnetic storm occurred. For the Kp Index, it shows mostly at 0 and 1 revealing no occurrence of geomagnetic storm.

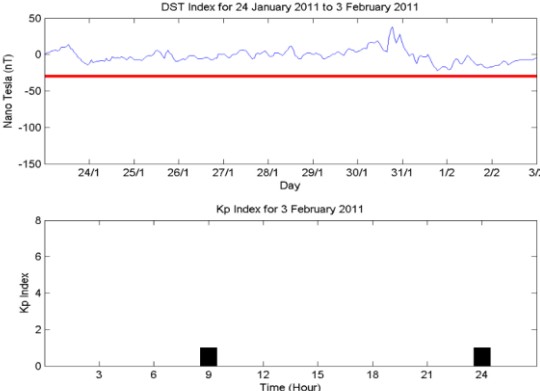

5    **Figure 3: Dst Index for 24 January to 3 February and Kp Index for normal period on 3 February, 2011.**

**2.4 Geomagnetic storm period and normal period**

By using this method, a comparison between active period and normal period for the stations in the SAA region, mid latitude region, as well as high latitude region will be done. Active period can be defined as a day from 1 UT until 24 UT where the existence of the geomagnetic storm is below – 30 nT. For normal period, it can be define as a day from 1 UT until 10    24 UT whereby the value is consistently above -30 nT indicating no geomagnetic storm occurrence.

For this study, the date chosen for analysis was on 3 February 2011 for a normal period and on 11 March 2011 for an active period as shown in Table 2. Year 2011 was chosen to study the SAA during the rising phase of solar cycle 24.

Table 2: Date to be analyzed.

| Active Period | Normal Period |
| --- | --- |
| 11 March 2011 | 03 February 2011 |

15   The date for active period, 11 March 2011 was selected since during that day, the geomagnetic storm is consistently below - 30 nT. Between 3 February 2011 and 11 March 2011, there are a number of geomagnetic storm occurrence. However, for the geomagnetic storm occurring on between those date, it can be seen the existing of the geomagnetic storm is inconsistent. For geomagnetic storm dates where it occurs consistently from 1 UT until 24 UT and closest to 3 February 2011, the date on which

the geomagnetic storm occurred was on 11 March 2011. On that date, 11 March 2011, the geomagnetic storm was found happens every time from 1 UT up to 24 UT according to Dst index.

The explanation on choosing 11 March 2011 compared to other date can be explained in more detail by referring Fig. 4. It can be seen through the Dst index, for example, on 1 March 2011, despite geomagnetic storm events but it is

inconsistent unlike the geomagnetic storm on 11 March 2011.

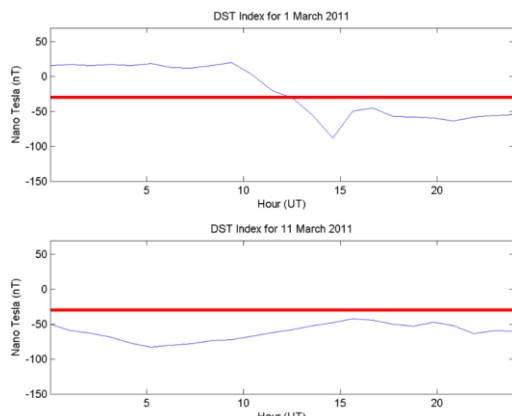

**Figure 4: Comparison between geomagnetic storm on 1 March 2011 and 11 March 2011.**

It is to be seen that on 1 March 2011, the geomagnetic storm started on that day at 13 UT up to 24 UT, but from 1 UT to 12 UT, no geomagnetic storm occurred. For the geomagnetic storm on 11 March 2011, it starts from 1 UT to 24 UT which is consistent throughout the day. The date of 11 March 2011 was also selected as it was closest to February 3, 2011.

**2.5 H-Component**

There are some characteristic that should be addressed regarding this work (Anwar et al., 2017). The abnormally large amplitude of the horizontal geomagnetic field component measured at the magnetic equator is caused by the intense current flowing in the equatorial ionosphere (Hamid et al., 2013). This variation can be observed by monitoring the geomagnetic field using the global network of magnetic observatories (Hamid et al., 2010). For this research, the component of the Earth's magnetic field chosen to be analyzed is the horizontal intensity (H), since it is additionally sensitive towards geomagnetic

activeness level. It can be studied through the beginning of a magnetic storm which is frequently described by a global sudden rise in H, that is mentioned as the storm sudden commencement or moreover expressed as SSC. Subsequent to the SSC, the H component normally remains on top of its average level for several hours. This stage is named as the initial phase of the storm.

Deleted: compare

Deleted: T

Afterwards, a great global decrease in H commences, signifying the evolvement of the main phase of the storm. Among the Earth's magnetic field component such as the total intensity (F), the inclination angle (I), the declination angle (D), the northerly intensity (X), the easterly intensity (Y) and the vertical intensity (Z), the horizontal intensity (H) is chosen due to this reason since in this research a comparison between a period when a geomagnetic storm occur and a period where no geomagnetic storm occur is conducted.

**3 Results and Discussion**

Figure 5 is an example of figure for power spectral density. The periodogram is obtained from station THL on 03 February 2011. The slope value is -1.6292. The value of spectral exponent, β is given by the negative slope of the straight line plot p (f) versus f in log-log scale known as the periodogram.

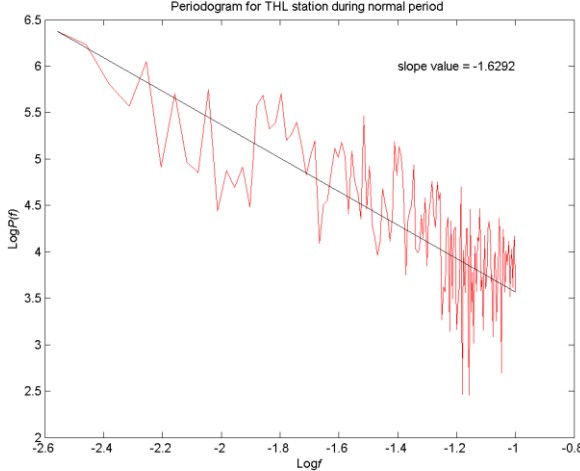

**Figure 5: An example of figure for power spectral density on 03 February 2011 station THL.**

Figure 6 show the periodogram for the high latitude region. The red periodogram represent the active period while the blue periodogram represent the normal period. The spectral exponent, β is in the range $1 < \beta \leq 3$. The spectral exponent, β is acquire through the negative value of the slope of the best-fit straight line corresponding to the selected frequency range. The spectral exponent, β will be apply in $H_{PS} = (\beta - 1)/2$. The Hurst exponent, H can determine the characteristic of the region.

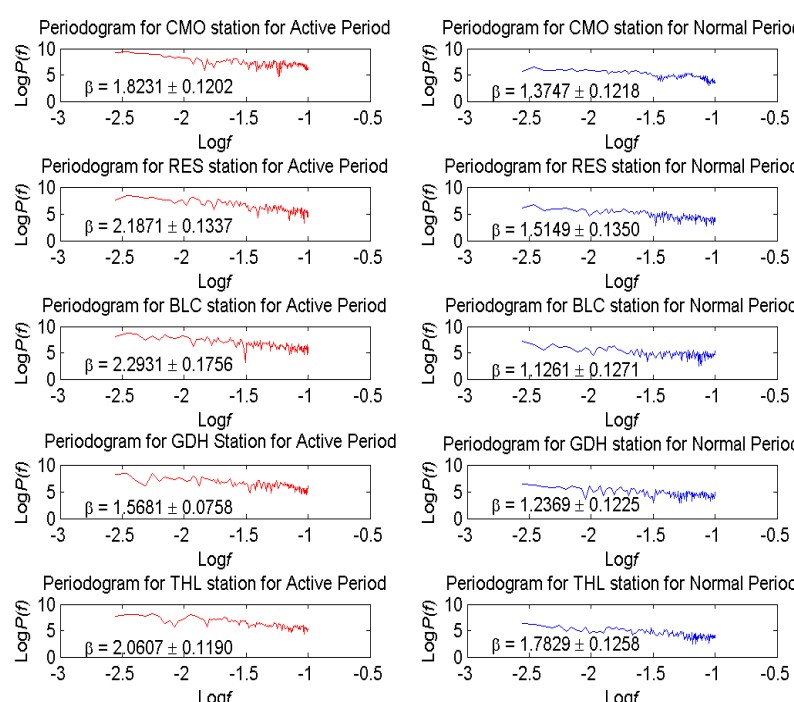

**Figure 6: Periodogram for high region during active period and normal period.**

Three regions were selected for this study; the first was high latitude region, second was mid latitude region, and third was the SAA region. A comparison between these regions was made to investigate the characteristics of the SAA region. Table 3, Table 4 and Table 5 show the research results. Table 3 shows the results for high latitude region during geomagnetic storm period (active period) and normal period, while Table 4 presents the results for mid latitude region during geomagnetic storm

5  (active period) and normal period. Table 5 represents the results of the SAA region. The maximum and minimum Earth's magnetic field strength for stations involved in the research are also shown in the tables during active and normal periods.

Table 3: Hurst exponent, the maximum and minimum Earth's magnetic field strength during geomagnetic storm (active period) and normal periods for stations in high latitude region.

| Station | Active Period | | | Normal Period | | |
|---|---|---|---|---|---|---|
| | The Hurst exponent value | Minimum Earth magnetic field strength (nT) | Maximum Earth magnetic field strength (nT) | The Hurst exponent value | Minimum Earth magnetic field strength (nT) | Maximum Earth magnetic field strength (nT) |
| CMO | 0.4116 ± 0.0601 | 56050 | 57060 | 0.1873 ± 0.0609 | 56910 | 56950 |
| GDH | 0.2840 ± 0.0379 | 56270 | 57160 | 0.1185 ± 0.0613 | 56500 | 56610 |
| THL | 0.5303 ± 0.0595 | 56360 | 56490 | 0.3915 ± 0.0629 | 56400 | 56410 |
| RES | 0.5936 ± 0.0669 | 57760 | 58010 | 0.2574 ± 0.0675 | 57820 | 57850 |
| BLC | 0.6466 ± 0.0878 | 58790 | 59440 | 0.0631 ± 0.0636 | 59090 | 59160 |

As for high latitude region, the stations chosen were from $60^0$ to $90^0$ latitude. It can be seen that the Hurst exponent value is varied. During active period, BLC, RES, THL stations showed persistent values with values of 0.6466 ± 0.0878, 0.5936 ± 0.0669 and 0.5303 ± 0.0595, respectively, while GDH and CMO stations showed antipersistent values of 0.2840 ± 0.0379 and 0.4116 ± 0.0601, respectively.

15  During the normal period, in the absence of a geomagnetic storm, the stations at high latitudes tend to record an antipersistent value. CMO, GDH, THL, RES and BLC stations values were 0.1873 ± 0.0609, 0.1185 ± 0.0613, 0.3915 ± 0.0629, 0.2574 ± 0.0675 and 0.0631 ± 0.0636, respectively. The obtained antipersistent values revealed a time series with

long-term switching between high and low values in adjacent pairs. It means a single high value may well be succeeded by a low value and the following value will tend to be high. This trend to alternate between high as well as low values will continue for a long time in the future.

It can be said the persistent and antipersistent experiences by stations in high latitude region may be associated with the strength of the Earth's magnetic field. The Earth's magnetic field value is strong in the high latitude area, thus providing a high degree of antipersistent for stations in the high latitude region. Minimum and maximum Earth's magnetic field values in the active period range from 56050 nT to 58790 nT and 57060 nT to 59440 nT, respectively. During normal period, the minimum and maximum Earth's magnetic field ranges from 56910 nT to 59090 nT and 56950 nT to 59160 nT, respectively.

It can be seen that Hurst exponent for station BLC is persistent while the Earth magnetic field strength is high. Should be based on the outcome of the result, the Hurst exponent of the station BLC is antipersistent when the Earth's magnetic field strength is high. This may be due to, for example, due to energetic particle factors. Station BLC is exposed to energetic particles especially when geomagnetic storm occurs during active period. Perhaps the energetic particles resulting from geomagnetic storms are able to affect the H component of the Earth magnetic field causing Hurst exponents in station BLC to produce persistent value. This is because the Earth magnetic field may change due to energetic particles originating from the geomagnetic storm. This is likely to be more concentrated in the BLC area that causes the station BLC to be affected in its Hurst exponent value.

Figure 7 represent the periodogram for mid latitude region during active period and normal period. The mid latitude region is situated in $30^0$ to $60^0$ latitude.

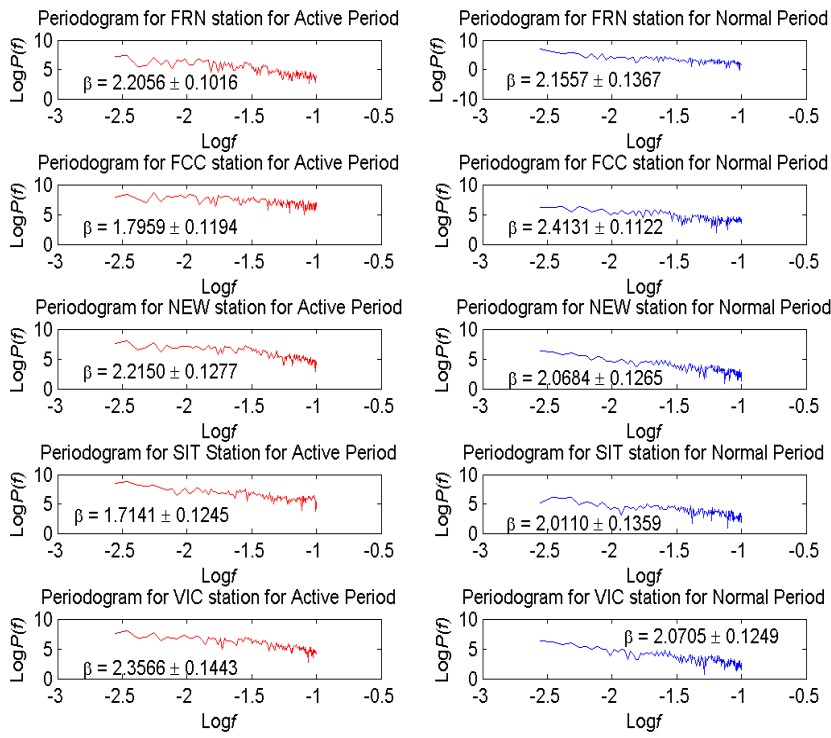

**Figure 7: Periodogram for mid latitude region during active period and normal**

For mid latitude region, table 4 shows the results of Hurst exponent value during geomagnetic storm (active period) and normal period.

Table 4: Hurst exponent, the maximum and minimum Earth's magnetic field strength during geomagnetic storm (active period) and normal period for station in mid latitude region.

| Station | Active Period | | | Normal Period | | |
|---|---|---|---|---|---|---|
| | The Hurst exponent value | Minimum Earth magnetic field strength (nT) | Maximum Earth magnetic field strength (nT) | The Hurst exponent value | Minimum Earth magnetic field strength (nT) | Maximum Earth magnetic field strength (nT) |
| FRN | 0.6028 ± 0.0508 | 48760 | 48800 | 0.5779 ± 0.0683 | 48780 | 48810 |
| VIC | 0.6783 ± 0.0722 | 54200 | 54330 | 0.5352 ± 0.0624 | 54270 | 54280 |
| NEW | 0.6075 ± 0.0639 | 55170 | 55320 | 0.5342 ± 0.0632 | 55250 | 55260 |
| SIT | 0.3571 ± 0.0622 | 55750 | 56320 | 0.5055 ± 0.0680 | 56050 | 56070 |
| FCC | 0.3980 ± 0.0597 | 58410 | 59260 | 0.7066 ± 0.0561 | 58940 | 58980 |

In the mid latitude regions, Earth's magnetic field values are not as strong as the high latitude region. The minimum and maximum values for the Earth's magnetic field during the active period were in the range of 48760 nT to 58410 nT and 48800 nT to 59260 nT, respectively. During normal periods, the minimum and maximum values of the Earth's magnetic field ranged from 48780 nT to 58940 nT and 48810 nT to 58980 nT, respectively. The Hurst exponent value showed a mixture of persistent and antipersistent values. SIT and FCC stations showed antipersistent values during active period. The value were 0.3571 ± 0.0622 and 0.3980 ± 0.0597. During normal period, the value was persistent with 0.5055 ± 0.0680 and 0.7066 ± 0.0561. As for VIC stations, NEW and FRN showed persistent values during active period and normal period.

It can be said that the value obtained in the mid latitude region differs from the value in the high latitude region during normal period and the active period may be due to different Earth's magnetic field values. Minimum Earth's magnetic field for mid latitude region during active period and normal period is from 48760 nT to 58410 nT and 48780 nT to 58940 nT, respectively. For the maximum Earth's magnetic field values were 48800 nT to 59260 nT during active period and 48810 nT to 58980 nT during normal period. This is in contrast to the high latitude region that has a higher Earth's magnetic field.

In addition to the Earth magnetic field, latitude position can also affect Hurst exponent. It can be seen that station NEW, VIC and FRN are nearer to the SAA region, thus affecting the value of Hurst Exponent to be persistent. This is because in the SAA region, the value of Hurst exponent tends to be persistent. For SIT and FCC stations are located far from the SAA region and closer to the high latitude region where the region is more vulnerable to be antipersistent.

Figure 8 indicate the periodogram for SAA region during active period as well as normal period.

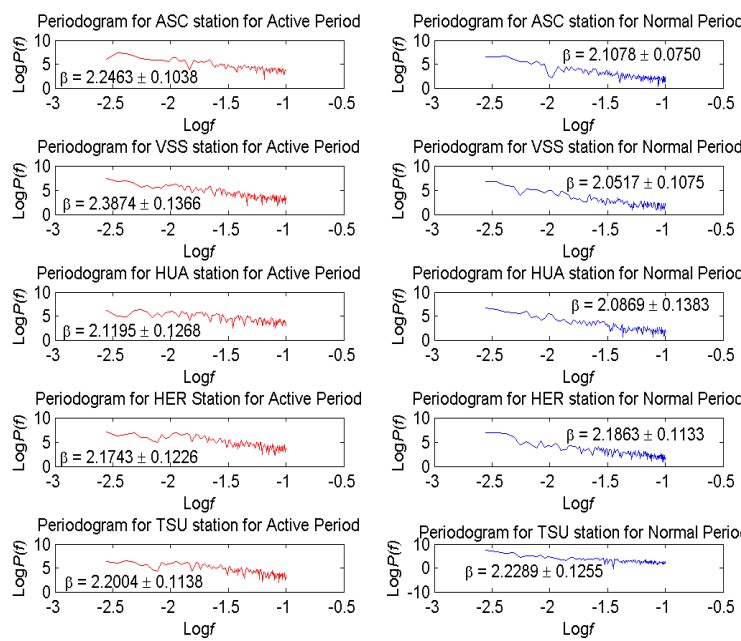

**Figure 8: Periodogram for SAA latitude region during active period and normal period.**

For SAA region, table 5 reveal the results of Hurst exponent value during geomagnetic storm (active period) and normal period.

Table 5: Hurst exponent, the maximum and minimum Earth's magnetic field strength during geomagnetic storm (active period) and normal period for station in SAA region.

| Station | Active Period | | | Normal Period | | |
|---|---|---|---|---|---|---|
| | The Hurst exponent value | Minimum Earth magnetic field strength (nT) | Maximum Earth magnetic field strength (nT) | The Hurst exponent value | Minimum Earth magnetic field strength (nT) | Maximum Earth magnetic field strength (nT) |
| VSS | 0.6937 ± 0.0683 | 23270 | 23350 | 0.5259 ± 0.0537 | 23340 | 23360 |
| HUA | 0.5597 ± 0.0634 | 25210 | 25470 | 0.5434 ± 0.0692 | 25320 | 25410 |
| HER | 0.5871 ± 0.0613 | 25730 | 25820 | 0.5932 ± 0.0567 | 25760 | 25810 |
| ASC | 0.6232 ± 0.0519 | 28270 | 28350 | 0.5539 ± 0.0375 | 28350 | 28380 |
| TSU | 0.6002 ± 0.0569 | 29600 | 29650 | 0.6144 ± 0.0627 | 29640 | 29670 |

In the SAA region, the value was indicated as persistent during active period and normal period. VSS, HUA, HER, ASC and TSU stations during the active period it was indicated as persistent with values of $0.6937 \pm 0.0683$, $0.5597 \pm 0.0634$, $0.5871 \pm 0.0613$, $0.6232 \pm 0.0519$ and $0.6002 \pm 0.0569$, respectively. During normal periods, VSS, HUA, HER, ASC and TSU stations recorded persistent values with Hurst exponent at $0.5259 \pm 0.0537$, $0.5434 \pm 0.0692$, $0.5932 \pm 0.0567$, $0.5539 \pm 0.0375$ and $0.6144 \pm 0.0627$, respectively.

The minimum strength of Earth's magnetic field in SAA region during active period and normal period ranged from 23270 nT to 29600 nT and 23340 nT to 29640 nT, respectively. As for maximum Earth's magnetic fields in the active period and normal period, the values were 23350 nT to 29650 nT and 23360 nT to 29670 nT, respectively. It could be possible to say that the SAA region experienced this characteristic because of the weak Earth's magnetic field it exhibited whereby high energy particles in the SAA region effecting the value of the H-component.

## 4 Conclusion

From the research conducted, observation made shows that the persistent and antipersistent values experienced by stations in the high latitude region and mid latitude region and the persistent tendency the SAA region experienced during active period and normal period could be due to the influence of the strength of the Earth's magnetic field. Regions that have a strong Earth's magnetic field is more indicative to be anti persistent. This can be seen in the high latitude area. As the Earth's magnetic field decreases, the more likely it is to appear persistent. This is happening in middle latitude region. Region where the Earth's magnetic field is very low, it shows a tendency for being persistent. This can be seen in the SAA region.

Certain factor such as station BLC in the high latitude region which reveal a persistent value even when the Earth magnetic field is high during active period may be due to the influence of energetic particle since it occur during geomagnetic storm. In addition, the position of a station can also affect the value of Hurst exponent. This can be seen at stations in mid latitude regions such as NEW, VIC and FRN which reveal a persistent value as it much closer to the SAA region. Similarly, stations such as FCC and SIT situated in mid latitude region, which is nearer to the high latitude region, display an antipersistent value.

The outcomes obtained from the research show that since the SAA region is persistent during active and normal periods in this study, meaning that a high value in the series will probably be followed by another high value, and that the values for a long time into the future will to be high. The possibility of a correlation can be made on the energetic particles in the SAA region whereby a large number of energetic particles in the SAA region can also be said to remain high in the future based on the tendency of the persistent value it experiences.

## Data Availability

Data for this research can be obtained from INTERMAGNET. It can be accessed at www.intermagnet.org.

## Author Contribution

Mardina Abdullah supervise the research and give approval as well as feedback. Nurul Shazana Abdul Hamid design the coding for power spectrum analysis and Hurst exponent and provide input on how the research should be done. Khairul Afifi Nasuddin do the research based on the supervision of Mardina Abdullah and Nurul Shazana Abdul Hamid and prepared the manuscript with contribution from all authors.

## Competing Interests

The authors declare that they have no conflict of interest.

**Acknowledgements**

The results presented in this paper rely on data collected at magnetic observatories. We thank the national institutes that support them and INTERMAGNET for promoting high standards of magnetic observatory practice (www.intermagnet.org).

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
