# Peer review of "Characterization of the South Atlantic Anomaly"

_Nonlinear Processes in Geophysics, 2018_

## Referee Comment (RC1) · Anonymous Referee #1 · 18 Dec 2018

article [utf8]inputenc indentfirst mwe

**1   General comments**

The South Atlantic Anomaly region is an important part in the earth.  To characterize the persistence in the SAA region is important and interesting.
The authors discuss the persistence in the SAA region and outside the SAA region during the active period and normal period, using power-spectral density analysis method. Their findings suggest that the SAA region tends to be persistent during both active and normal period. The persistence may be explained by the earth magnetic field.
The aim, the method and the motivation lie within the nonlinear progress in Geogphysics.
However, there are some deficiencies.

1. I suggest add an example figure of power spectral density in order to see in which frequency range you estimate the slope and whether the oscillations affect the slope estimation. An exact estimation of the slope is quite important to the results and conclusions. To me, the Hurst value=0.06 is really strange.

2. There are still stations which are against your conclusion, like SIT and FCC station. As I see, the results of these two stations are totally different from your conclusion. I suggest add more explanation about that.

**2   Specific comments**

**2.1   Abstract**

Line 4, "the data for the occurrence of the active period and normal period. . .". The data of which variable is not mentioned in the abstract part. And which temporal resolution of the data are you using?

**2.2   Methodology**

$\beta$ is estimated by the slope of the power-spectral density function, it must be valid in a certain frequency range. So it is better to present an example figure of a power-spectral density of one station. Then it is much clearer to see how you estimate the slope, which frequency range it is and if the oscillations influence the estimation of the $\beta$. Like Figure 2 in Shao and Ditlevsen., 2015, Figure 1-3 in Pelletier and Turcotte., 1997.

Sentence: "If the Hurst exponent is in the range of 0.5-1, it reveals a time series with long-term positive autocorrelation". "In the range of 0.5-1" is not accurate. Does H=0.5

fall into the range of 0.5-1? Change it to 0.5<H<1.
"a long-term positive autocorrelation" is not accurate, either. Check it.
Add the explanation of H=0.5.

**2.3   Dst index and Kp index for geomagnetic storms**

This part should be put in front of the 2.3 part. In 2.3 you have already applied the Dst index to choose the active and normal period.
The title of Figure 3(1), the xlabel means day 1 March to 16 March, not for 11 March.
Figure 3(2), xlabel "Time" means "Time/hour"?
Also check the title of Figure 4(1).

**2.4   H-component**

The temporal resolution of the data you used is not mentioned in the paper, is it minutely data?
From Figure 5, I can see a diurnal cycle of horizontal intensity (H). Oscillations in the data will lead a peak (See Shao and Ditlevsen., 2016). The peak may influence the Hurst exponent. You can provide an example of power-spectral density and see if the peak fall into the range of slope estimation. If it really affects the slope, you need to remove the diurnal cycle and re-estimate the Hurst exponent.
Line 7, the ssc should be capital letters.

**2.5   Results and Discussion**

For BLC station, during active period, the minimum Earth magnetic field strength is 58790, but the H(BLC)=0.6466. For SIT station, the minimum Earth magnetic field strength is 55750, but the H(SIT)=0.3517. It is completely different from your result.

I suggest add an explanation about that. And the Hurst exponents of SIT and FCC stations during active period are lower than them during the normal period, which are totally different from other stations. Also an explanation for it.

**2.6   Conclusion**

Line 5, 'This is happening in low latitude region.' You mean the mid latitude region?

**2.6.1   Typos**

Page 7, line 26,'The explanation on choosing 11 March 2011 compare to other date can be explain more detail. . .' should be 'be explained'.

**3   References**

Shao, Z. G., & Ditlevsen, P. D. (2016). Contrasting scaling properties of interglacial and glacial climates. Nature communications, 7, 10951.
Pelletier, J. D., & Turcotte, D. L. (1997). Long-range persistence in climatological and hydrological time series: analysis, modeling and application to drought hazard assessment. Journal of Hydrology, 203(1-4), 198-208.

---

## Referee Comment (RC2) · Anonymous Referee #2 · 20 Dec 2018

The paper can not recommended to publish in the present form, due to the following reasons, 1.Since it is long-range correlation analysis, there should be the plots showing how to fit the scaling exponent, which range has been used. This kind of important figures are not given in the manuscript. The conclusions given in this manuscript may be not believable.

2. In the Fig. 5, there are changing cycles, which will distort the estimation of spectal exponent \beta, how do you deal with them? This should be explain in the manuscript.

3. The strength of the Earth's magnetic field influences the estimated spectal exponent \beta over different regions and different phases can not explain all the results given in Tables 3-5, exceptional results for some specific stations should be explained.

4. The data length used in this study. Is there any finite effect on the estimated spectal exponent?

[Figure]

5. If there are mutiple scaling ranges, which one is chosen to fit estimated spectal exponent? Why?

---

## Author Comment (AC1) · 28 Jan 2019

Response to referee comments

Manuscript title: Characterization of the South Atlantic Anomaly

Authors: Khairul Afifi Nasuddin, Mardina Abdullah, Nurul Shazana Abdul Hamid

We like to thank the referee for their comment. We have read and prepare the respond. The summarization of the respond is explain in this attachment.

Referee comment 1

1. General comments

1. I suggest add an example figure of power spectral density in order to see in which frequency range you estimate the slope and whether the oscillations affect the

[Discussion paper]

[Figure]

slope estimation. An exact estimation of the slope is quite important to the results and conclusions. To me, the Hurst value=0.06 is really strange.

Response: We have inserted the periodogram figure in the Results and Discussion section in line 8 page 11 figure 6 for the high latitude region, the periodogram figure for mid latitude region in line 3 page 14 figure 7 and the periodogram for South Atlantic Anomaly region in line 7 page 16 figure 8. The spectral exponent, $\beta$ for the station in the region can be seen the periodogram figure. An example figure of power spectral density is shown in page 10 line 7 figure 5.

2. There are still stations which are against your conclusion, like SIT and FCC station. As I see, the results of these two stations are totally different from your conclusion. I suggest add more explanation about that.

Response: The explanation have been added in the conclusion section line 1 to line 4 page 16 and line 11 to line 14 in page 18.

2. Specific comments

2.1 Abstract Line 4, " the data for the occurrence of the active period and normal period... ". The data of which variable is not mentioned in the abstract part. And which temporal resolution of the data are you using?

Response: We have mention the horizontal-component of the Earth magnetic field as the component to be analyze and data sample rate is 1 minute in line 11 and 12 in page 1 in the Abstract section.

2.2 Methodology

$\beta$ is estimated by the slope of the power-spectral density function, it must be valid in a certain frequency range. So it is better to present an example figure of a power spectral density of one station. Then it is much clearer to see how you estimate the slope, which frequency range it is and if the oscillations influence the estimation of the $\beta$. Like Figure 2 in Shao and Ditlevsen., 2015, Figure 1-3 in Pelletier and Turcotte.,

1997.

Response: The spectral exponent, $\beta$ for the station in the region can be seen the periodogram figure. The periodogram figure is inserted in the Results and Discussion section in line 8 page 11 figure 6 for the high latitude region, the periodogram figure for mid latitude region in line 3 page 14 figure 7 and the periodogram for South Atlantic Anomaly region in line 7 page 16 figure 8. The spectral exponent, $\beta$ for the station in the region can be seen the periodogram figure. An example figure of power spectral density of one station is shown in page 10 line 7 figure 5.

Sentence: "If the Hurst exponent is in the range of 0.5-1, it reveals a time series with long-term positive autocorrelation". "In the range of 0.5-1" is not accurate. Does H=0.5 fall into the range of 0.5-1? Change it to 0.5<H<1.

Response: We have change it. It is in line 5 page 7.

"a long-term positive autocorrelation" is not accurate, either. Check it.

Response: We have improve the sentence. It is in line 5 page 7.

Add the explanation of H=0.5.

Response: We have add explanation on H = 0.5. It is line 9 and 10 in page 7.

2.3 Dst index and Kp index for geomagnetic storms

This part should be put in front of the 2.3 part. In 2.3 you have already applied the Dst index to choose the active and normal period.

Response: We have place it as suggested by the referee. Section 2.3 Dst index and Kp index for geomagnetic storm period and normal period have been removed to page 7 and section 2.4 Geomagnetic storm period and normal period has been place in page 8.

The title of Figure 3(1), the x label means day 1 March to 16 March, not for 11 March.

Figure 3(2), x label "Time" means "Time/hour"? Also check the title of Figure 4(1).

Response: We have corrected the mistake. Figure 3(1) and figure 3(2) have been corrected based on suggestion. It has been rename as Figure 2 since the arrangement of section 2.3 Dst index and Kp index for geomagnetic storm period and normal period with section 2.4 Geomagnetic storm period and normal period. The correction is in page 7. Figure 3(1) and Figure 3(2) (Previously name figure 4) also been corrected in page 8.

2.4 H-component

The temporal resolution of the data you used is not mentioned in the paper, is it minutely data? From Figure 5, I can see a diurnal cycle of horizontal intensity (H). Oscillations in the data will lead a peak (See Shao and Ditlevsen., 2016). The peak may inïñĆuence the Hurst exponent. You can provide an example of power-spectral density and see if the peak fall into the range of slope estimation. If it really affects the slope, you need to remove the diurnal cycle and re-estimate the Hurst exponent. Line 7, the ssc should be capital letters.

Response: The data sample rate is 1-minute. An example figure of power spectral density in which frequency range whereby the slope is estimate is shown in page 10 line 7 figure 5. We have change ssc to SSC.

2.5 Results and Discussion

For BLC station, during active period, the minimum Earth magnetic ïñĄeld strength is 58790, but the H(BLC)=0.6466. For SIT station, the minimum Earth magnetic ïñĄeld strength is 55750, but the H(SIT)=0.3517. It is completely different from your result. I suggest add an explanation about that. And the Hurst exponents of SIT and FCC stations during active period are lower than them during the normal period, which are totally different from other stations. Also an explanation for it.

Response: The explanation is mention in the results and discussion section for station

BLC in section line 12 to line 19 in page 13. The explanation on SIT and FCC is mention in the line 1 to line 4 page 16.

2.6 Conclusion

Line 5, 'This is happening in low latitude region.' You mean the mid latitude region?

Response: We have corrected the mistake. It is in line 5 page 18.

2.6.1 Typos

Page 7, line 26,' The explanation on choosing 11 March 2011 compare to other date can be explain more detail...' should be ' be explained '.

Response: We have change the sentence to be explained. It is in line 3 page 9.

Please also note the supplement to this comment:
https://www.nonlin-processes-geophys-discuss.net/npg-2018-51/npg-2018-51-AC1-supplement.pdf

**Supplement:**

[revised manuscript text omitted]

---

## Author Comment (AC2) · 28 Jan 2019

Response to referee comments

Manuscript title: Characterization of the South Atlantic Anomaly

Authors: Khairul Afifi Nasuddin, Mardina Abdullah, Nurul Shazana Abdul Hamid

We like to thank the referee for their comment. We have read and prepare the respond. The summarization of the respond is explain in this attachment.

Referee comment 2

Since it is long-range correlation analysis, there should be the plots showing how to iňĄt the scaling exponent, which range has been used. This kind of important iňĄgures are not given in the manuscript. The conclusions given in this manuscript may be not
believable.

Response: The periodogram figure can be seen in the Results and Discussion section in line 8 page 11 figure 6 for the high latitude region, the periodogram figure for mid latitude region in line 3 page 14 figure 7 and the periodogram for South Atlantic Anomaly region in line 7 page 16 figure 8. The spectral exponent,  $\beta$  for the station in the region can be seen the periodogram figure. An example figure of power spectral density is shown in page 10 line 7 figure 5.

In the Fig. 5, there are changing cycles, which will distort the estimation of spectal exponent\beta, how do you deal with them? This should be explain in the manuscript.

Response: For the 2.4 H-component, the purpose of this section is to explain the selection of the H-component apply in this research compare to other Earth's magnetic field component. An example figure of power spectral density is shown in page 10 line 7 figure 5.

The strength of the Earth's magnetic ïňĄeld inïňĆuences the estimated spectral exponent \beta over different regions and different phases cannot explain all the results given in Tables 3-5, exceptional results for some speciïňĄc stations should be explained.

Response: The explanation have been explain in the conclusion section. It is in line 9 to line 14 page 18.

The data length used in this study. Is there any *ïň*Anite effect on the estimated spectral exponent?

Response: The data length is for 1 day which is 24 hour. A comparison is made between data of the H-component for 1 day between the occurrence of geomagnetic storm and when no geomagnetic storm occur. By applying the power spectrum analysis method, the spectral exponent  $\beta$  can be obtained. Thus, the Hurst exponent can be define.

NPGD
If there are multiple scaling ranges, which one is chosen to inAt estimated spectral exponent? Why?

Response: The method chosen to analyze the South Atlantic Anomaly is power spectrum analysis. Power spectrum analysis is a representation of the magnitud of the various frequency components of a signal. By looking at the spectrum, one can find how much energy or power is contained in the frequency components of the signal. The one scaling range is chosen since the data sample rate is 1 minute.

Please also note the supplement to this comment:

https://www.nonlin-processes-geophys-discuss.net/npg-2018-51/npg-2018-51-AC2-supplement.pdf

---

## Author Response (AR1)

**Response to referee comments**

**Manuscript title:** Characterization of the South Atlantic Anomaly

**Authors:** Khairul Afifi Nasuddin, Mardina Abdullah, Nurul Shazana Abdul Hamid

We like to thank the referee for their comment. We have read and prepare the respond. The summarization of the respond is explain in this attachment.

**Referee comment 1**

**1. General comments**

**(1) Comments from Referees:** *1. I suggest add an example figure of power spectral density in order to see in which frequency range you estimate the slope and whether the oscillations affect the slope estimation. An exact estimation of the slope is quite important to the results and conclusions. To me, the Hurst value=0.06 is really strange.*

**(2) Author's response:** An example figure of power spectral density is shown in page 10 line 7 figure 5. We also have inserted the periodogram figure in the Results and Discussion section in line 8 page 11 figure 6 for the high latitude region, the periodogram figure for mid latitude region in line 3 page 14 figure 7 and the periodogram for South Atlantic Anomaly region in line 7 page 16 figure 8. The spectral exponent, $\beta$ for the station in the region can be seen the periodogram figure.

**(1) Comments from Referees:** *2. There are still stations which are against your conclusion, like SIT and FCC station. As I see, the results of these two stations are totally different from your conclusion. I suggest add more explanation about that.*

**(2) Author's response:** The explanation have been added in the results and discussion section line 1 to line 4 page 16 and in the conclusion section line 11 to line 14 in page 18.

**(3) Author's changes:**

Results and discussion section line 1 to line 4 page 16 :-

In addition to the Earth magnetic field, latitude position can also affect Hurst exponent. It can be seen that station NEW, VIC and FRN are nearer to the SAA region, thus affecting the value of Hurst Exponent to be persistent. This is because in the SAA region, the value of Hurst exponent tends to be persistent. For SIT and FCC stations are located far from the SAA region and closer to the high latitude region where the region is more vulnerable to be antipersistent.

Conclusion section line 11 to line 14 in page 18 :-

In addition, the position of a station can also affect the value of Hurst exponent. This can be seen at stations in mid latitude regions such as NEW, VIC and FRN which reveal a persistent value as it much closer to the SAA region. Similarly, stations such as FCC and SIT situated in mid latitude region, which is nearer to the high latitude region, display an antipersistent value.

**2. Specific comments**

**2.1 Abstract**

**(1) Comments from Referees:** Line 4, " the data for the occurrence of the active period and normal period... ". The data of which variable is not mentioned in the abstract part. And which temporal resolution of the data are you using?

**(2) Author's response:** We have mention the horizontal-component of the Earth magnetic field as the component to be analyze and data sample rate is 1 minute in line 11 and 12 in page 1 in the Abstract section.

**(3) Author's changes:**

The horizontal component of the Earth magnetic field data for the occurrence of the active period was taken on 11 March 2011 while for normal period on 3 February 2011. The used data sample rate is 1-minute.

**2.2 Methodology**

**(1) Comments from Referees:** $\beta$ is estimated by the slope of the power-spectral density function, it must be valid in a certain frequency range. So it is better to present an example figure of a power spectral density of one station. Then it is much clearer to see how you estimate the slope, which frequency range it is and if the oscillations influence the estimation of the β. Like Figure 2 in Shao and Ditlevsen., 2015, Figure 1-3 in Pelletier and Turcotte., 1997.

**(2) Author's response:** The spectral exponent, $\beta$ for the station in the region can be seen the periodogram figure. The periodogram figure is inserted in the Results and Discussion section in line 8 page 11 figure 6 for the high latitude region, the periodogram figure for mid latitude region in line 3 page 14 figure 7 and the periodogram for South Atlantic Anomaly region in line 7 page 16 figure 8. The spectral exponent, $\beta$ for the station in the region can be seen the periodogram figure. An example figure of power spectral density of one station is shown in page 10 line 7 figure 5.

**(1) Comments from Referees:** Sentence: "If the Hurst exponent is in the range of 0.5-1, it reveals a time series with long-term positive autocorrelation". "In the range of 0.5-1" is not accurate. Does H=0.5 fall into the range of 0.5-1? Change it to 0.5<H<1.

**(2) Author's response:** We have change it. It is in line 5 page 7.

**(3) Author's changes:**

If the Hurst exponent is in the range of 0.5<H<1, it can be interpret as both that a high value in the series will probably be followed by another high value also that the values a long time into the future will serve to be high.

**(1) Comments from Referees:** "a long-term positive autocorrelation" is not accurate, either. Check it.

**(2) Author's response:** We have improve the sentence. It is in line 5 page 7.

**(3) Author's changes:**

If the Hurst exponent is in the range of 0.5<H<1, it can be interpret as both that a high value in the series will probably be followed by another high value also that the values a long time into the future will serve to be high.

**(1) Comments from Referees:** Add the explanation of H=0.5.

**(2) Author's response:** We have add explanation on H = 0.5. It is line 9 and 10 in page 7.

**(3) Author's changes:**

While for H = 0.5 implies a random series. It can also mean data is not correlated, that is no dependence between current and past data.

**2.3 Dst index and Kp index for geomagnetic storms**

**(1) Comments from Referees:** This part should be put in front of the 2.3 part. In 2.3 you have already applied the Dst index to choose the active and normal period.

**(2) Author's response:** We have place it as suggested by the referee. Section 2.3 Dst index and Kp index for geomagnetic storm period and normal period have been removed to page 7 and section 2.4 Geomagnetic storm period and normal period has been place in page 8.

**(1) Comments from Referees:** The title of Figure 3(1), the x label means day 1 March to 16 March, not for 11 March. Figure 3(2), x label "Time" means "Time/hour"? Also check the title of Figure 4(1).

**(2) Author's response:** We have corrected the mistake. Figure 3(1) and figure 3(2) have been corrected based on suggestion. It has been rename as Figure 2 since the arrangement of section 2.3  Dst index and Kp index for geomagnetic storm period and normal period with section 2.4 Geomagnetic storm period and normal period. The correction is in page 7. Figure 3(1) and Figure 3(2) (Previously name figure 4) also been corrected in page 8.

**2.4 H-component**

**(1) Comments from Referees:** The temporal resolution of the data you used is not mentioned in the paper, is it minutely data? From Figure 5, I can see a diurnal cycle of horizontal intensity (H). Oscillations in the data will lead a peak (See Shao and Ditlevsen., 2016). The peak may influence the Hurst exponent. You can provide an example of power-spectral density and see if the peak fall into the range of slope estimation. If it really affects the slope, you need to remove the diurnal cycle and re-estimate the Hurst exponent. Line 7, the ssc should be capital letters.

**(2) Author's response:** The data sample rate is 1-minute. An example figure of power spectral density in which frequency range whereby the slope is estimate is shown in page 10 line 7 figure 5. We have change ssc to SSC.

**(3) Author's changes:**

Figure 5 is an example of figure for power spectral density. The periodogram is on 03 February 2011 (Normal period) station THL. The slope value is -1.6292. The value of spectral exponent, $\beta$ is given by the negative slope of the straight line plot p (f) versus f in log-log scale known as the periodogram.

[Figure]

**Figure 5: An example of figure for power spectral density on 03 February 2011 station THL.**

**2.5 Results and Discussion**

**(1) Comments from Referees:** For BLC station, during active period, the minimum Earth magnetic field strength is 58790, but the H(BLC)=0.6466. For SIT station, the minimum Earth magnetic field strength is 55750, but the H(SIT)=0.3517. It is completely different from your result.

I suggest add an explanation about that. And the Hurst exponents of SIT and FCC stations during active period are lower than them during the normal period, which are totally different from other stations. Also an explanation for it.

**(2) Author's response:** The explanation is mention in the results and discussion section for station BLC in line 12 to line 19 in page 13. The explanation on SIT and FCC is mention in the line 1 to line 4 page 16.

**(3) Author's changes**:

Results and discussion section line 12 to line 19 in page 13:-

It can be seen that Hurst exponent for station BLC is persistent while the Earth magnetic field strength is high. Should be based on the outcome of the result, the Hurst exponent of the station BLC is antipersistent when the Earth's magnetic field strength is high. This may be due to, for example, due to energetic particle factors. Station BLC is exposed to energetic particles especially when geomagnetic storm occurs during active period. Perhaps the energetic particles resulting from geomagnetic storms are able to affect the H component of the Earth magnetic field causing Hurst exponents in station BLC to produce persistent value. This is because the Earth magnetic field may change due to energetic particles originating from the geomagnetic storm. This is likely to be more concentrated in the BLC area that causes the station BLC to be affected in its Hurst exponent value.

Results and discussion section line 1 to line 4 in page 16:-

In addition to the Earth magnetic field, latitude position can also affect Hurst exponent. It can be seen that station NEW, VIC and FRN are nearer to the SAA region, thus affecting the value of Hurst Exponent to be persistent. This is because in the SAA region, the value of Hurst exponent tends to be persistent. For SIT and FCC stations are located far from the SAA region and closer to the high latitude region where the region is more vulnerable to be antipersistent.

**2.6 Conclusion**

**(1) Comments from Referees:** Line 5, 'This is happening in low latitude region.' You mean the mid latitude region?

**(2) Author's response:** We have corrected the mistake. It is in line 5 page 18.

**(3) Author's changes**:

This is happening in middle latitude region.

**2.6.1 Typos**

**(1) Comments from Referees:** Page 7, line 26,' The explanation on choosing 11 March 2011 compare to other date can be explain more detail...' should be ' be explained '.

**(2) Author's response:** We have change the sentence to be explained. It is in line 3 page 9.

**(3) Author's changes:**

The explanation on choosing 11 March 2011 compare to other date  can be explained more detail by referring Fig. 4.

**Referee comment 2**

**(1) Comments from Referees:** 1. Since it is long-range correlation analysis, there should be the plots showing how to fit the scaling exponent, which range has been used. This kind of important figures are not given in the manuscript. The conclusions given in this manuscript may be not believable.

**(2) Author's response: T**he periodogram figure can be seen in the Results and Discussion section in line 8 page 11 figure 6 for the high latitude region, the periodogram figure for mid latitude region in line 3 page 14 figure 7 and the periodogram for South Atlantic Anomaly region in line 7 page 16 figure 8. The spectral exponent, $\beta$ for the station in the region can be seen the periodogram figure. An example figure of power spectral density is shown in page 10 line 7 figure 5.

**(1) Comments from Referees:** 2. In the Fig. 5, there are changing cycles, which will distort the estimation of spectal exponent\beta, how do you deal with them? This should be explain in the manuscript.

**(2) Author's response:** For the 2.4 H-component, the purpose of this section is to explain the selection of the H-component apply in this research compare to other Earth's magnetic field component. An example figure of power spectral density is shown in page 10 line 7 figure 5.

**(3) Author's changes:**

Figure 5 is an example of figure for power spectral density. The periodogram is on 03 February 2011 (Normal period) station THL. The slope value is -1.6292. The value of spectral exponent, $\beta$ is given by the negative slope of the straight line plot p (f) versus f in log-log scale known as the periodogram.

[Figure]

**Figure 5: An example of figure for power spectral density on 03 February 2011 station THL.**

**(1) Comments from Referees:** 3. The strength of the Earth's magnetic field influences the estimated spectral exponent \beta over different regions and different phases cannot explain all the results given in Tables 3-5, exceptional results for some specific stations should be explained.

**(2) Author's response:** The explanation have been explain in the conclusion section. It is in line 9 to line 14 page 18.

**(3) Author's changes:**

Certain factor such as station BLC in the high latitude region which reveal a persistent value even when the Earth magnetic field is high during active period may be due to the influence of energetic particle since it occur during geomagnetic storm. In addition, the position of a station can also affect the value of Hurst exponent. This can be seen at stations in mid latitude regions such as NEW, VIC and FRN which reveal a persistent value as it much closer to the SAA region. Similarly, stations such as FCC and SIT situated in mid latitude region, which is nearer to the high latitude region, display an antipersistent value.

**(1) Comments from Referees:** 4. The data length used in this study. Is there any finite effect on the estimated spectral exponent?

**(2) Author's response:** The data length is for 1 day which is 24 hour. A comparison is made between data of the H-component for 1 day between the occurrence of geomagnetic storm and when no geomagnetic storm occur. By applying the power spectrum analysis method, the spectral exponent $\beta$ can be obtained. Thus, the Hurst exponent can be define.

**(1) Comments from Referees:** 5. If there are multiple scaling ranges, which one is chosen to fit estimated spectral exponent? Why?

**(2) Author's response:** The method chosen to analyze the South Atlantic Anomaly is power spectrum analysis. Power spectrum analysis is a representation of the magnitud of the various frequency components of a signal. By looking at the spectrum, one can find how much energy or power is contained in the frequency components of the signal. The one scaling range is chosen since the data sample rate is 1 minute.

[revised manuscript text omitted]
". "In the range of 0.5-1" is not accurate. Does H=0.5 fall into the range of 0.5-1? Change it to 0.5<H<1.

"a long-term positive autocorrelation" is not accurate, either. Check it.

**Commented [U3]:**
*Referee comment 1*
**2. Specific comments**
**2.2 Methodology**

Add the explanation of H=0.5.

**Commented [U4]:**
*Referee comment 1*
**2. Specific comments**
**2.3 Dst index and Kp index for geomagnetic storms**

This part should be put in front of the 2.3 part. In 2.3 you have already applied the Dst index to choose the active and normal period.

**Commented [U5]:**
*Referee comment 1*
**2. Specific comments**
**2.3 Dst index and Kp index for geomagnetic storms**

The title of Figure 3(1), the x label means day 1 March to 16 March, not for 11 March. Figure 3(2), x label "Time" means "Time/hour"? Also check the title of Figure 4(1).

For the normal period, on 3 February in 2011, the Dst Index indicate no geomagnetic storm occurred. For the Kp Index, it shows mostly at 0 and 1 revealing no occurrence of geomagnetic storm.

[Figure]

5    **Figure 3: Dst Index for 24 January to 3 February and Kp Index for normal period on 3 February, 2011.**

**Commented [U6]:**
*Referee comment 1*
**2. Specific comments**
**2.3 Dst index and Kp index for geomagnetic storms**

The title of Figure 3(1), the x label means day 1 March to 16 March, not for 11 March. Figure 3(2), x label "Time" means "Time/hour"? Also check the title of Figure 4(1).

**2.4 Geomagnetic storm period and normal period**

By using this method, a comparison between active period and normal period for the stations in the SAA region, mid latitude region, as well as high latitude region will be done. Active period can be define as a day from 1 UT until 24 UT where the existence of the geomagnetic storm is below – 30 nT. For normal period, it can be define as a day from 1 UT until 24 UT
10   whereby the value is consistently above -30 nT indicating no geomagnetic storm occurrence.

For this study, the date chosen for analysis was on 3 February 2011 for a normal period and on 11 March 2011 for an active period as shown in Table 2. Year 2011 was chosen to study the SAA during the rising phase of solar cycle 24.

Table 2: Date to be analyzed.

| Active Period | Normal Period |
|---|---|
| 11 March 2011 | 03 February 2011 |

15   The date for active period, 11 March 2011 was selected since during that day, the geomagnetic storm is consistently below - 30 nT. Between 3 February 2011 and 11 March 2011, there are a number of geomagnetic storm occurrence. However, for the geomagnetic storm occurring on between those date, it can be seen the existing of the geomagnetic storm is inconsistent. For geomagnetic storm dates where it occurs consistently from 1 UT until 24 UT and closest to 3 February 2011, the date on which

the geomagnetic storm occurred was on 11 March 2011. On that date, 11 March 2011, the geomagnetic storm was found happens every time from 1 UT up to 24 UT according to Dst index.

The explanation on choosing 11 March 2011 compare to other date  can be explained more detail by referring Fig. 4. It can be seen through the Dst index, for example, on 1 March 2011, despite geomagnetic storm events but it is inconsistent unlike the geomagnetic storm on 11 March  2011.

**Commented [U7]:** *Referee comment 1*
**2. Specific comments**
**2.6.1 Typos**

Page 7, line 26,' The explanation on choosing 11 March 2011 compare to other date can be explain more detail...' should be ' be explained '.

[Figure]

**Figure 4: Comparison between geomagnetic storm on 1 March 2011 and 11 March 2011.**

It is to be seen that on 1 March 2011, the geomagnetic storm started on that day at 13 UT up to 24 UT, but from 1 UT to 12 UT,  no geomagnetic storm occurred. For the geomagnetic storm on 11 March 2011, it starts from 1 UT to 24 UT which is consistent throughout the day. The date of 11 March 2011 was also selected as it was closest to February 3, 2011.

**2.5 H-Component**

The component of the Earth's magnetic field chosen to be analyzed is the horizontal intensity (H), since it is additionally sensitive towards geomagnetic activeness level. It can be studied through the beginning of a magnetic storm which is frequently described by a global sudden rise in H, that is mentioned as the storm sudden commencement or moreover expressed as SSC. Subsequent to the SSC, the H component normally remains on top of its average level for several hours. This stage is named as the initial phase of the storm. Afterwards, a great global decrease in H commences, signifying the evolvement of the main phase of the storm. Among the Earth's magnetic field component such as the total intensity (F), the inclination angle (I), the declination angle (D), the northerly intensity (X), the easterly intensity (Y) and the vertical intensity

**Commented [U8]:**
*Referee comment 1*
**2. Specific comments**
**2.4 H-component**

The ssc should be capital letters.

**Commented [U9]:**

*Referee comment 1*
**2. Specific comments**
**2.4 H-component**

The ssc should be capital letters.

(Z), the horizontal intensity (H) is chosen due to this reason since in this research a comparison between a period when a geomagnetic storm occur and a period where no geomagnetic storm occur is conducted.

**3 Results and Discussion**

Figure 5 is an example of figure for power spectral density. The periodogram is on 03 February 2011 (Normal period)
5   station THL. The slope value is -1.6292. The value of spectral exponent, β is given by the negative slope of the straight line plot p (f) versus f in log-log scale known as the periodogram.

[Figure]

10   **Figure 5: An example of figure for power spectral density on 03 February 2011 station THL.**

**Commented [U10]:**
*Referee comment 1*
*1. General Comment*
   *1.I suggest add an example figure of power spectral density in order to see in which frequency range you estimate the slope and whether the oscillations affect the slope estimation. An exact estimation of the slope is quite important to the results and conclusions. To me, the Hurst value=0.06 is really strange.*

**2. Specific comments**
**2.2 Methodology**
*β* is estimated by the slope of the power-spectral density function, it must be valid in a certain frequency range. So it is better to present an example figure of a power spectral density of one station. Then it is much clearer to see how you estimate the slope, which frequency range it is and if the oscillations influence the estimation of the β. Like Figure 2 in Shao and Ditlevsen., 2015, Figure 1-3 in Pelletier and Turcotte., 1997.

**Referee comment 2**
   1.Since it is long-range correlation analysis, there should be the plots showing how to fit the scaling exponent, which range has been used. This kind of important figures are not given in the manuscript. The conclusions given in this manuscript may be not believable.

**Commented [U11]:**
*Referee comment 1*
**2. Specific comments**
**2.4 H-component**

The temporal resolution of the data you used is not mentioned in the paper, is it minutely data? From Figure 5, I can see a diurnal cycle of horizontal intensity (H). Oscillations in the data will lead a peak (See Shao and Ditlevsen., 2016). The peak may influence the Hurst exponent. You can provide an example of power-spectral density and see if the peak fall into the range of slope estimation. If it really affects the slope, you need to remove the diurnal cycle and re-estimate the Hurst exponent. Line 7, the ssc should be capital letters.

**Referee comment 2**

2. In the Fig. 5, there are changing cycles, which will distort the estimation of spectal exponent\beta, how do you deal with them? This should be explain in the manuscript.

Figure 6 show the periodogram for the high latitude region. The red periodogram represent the active period while the blue periodogram represent the normal period. The spectral exponent, β is in the range $1 < \beta \leq 3$. The spectral exponent, β is acquire through the negative value of the slope of the best-fit straight line corresponding to the selected frequency range. The spectral exponent, β will be apply in $H_{PS} = (\beta - 1)/2$. The Hurst exponent, H can determine the characteristic of the region.

[Figure]

**Commented [U12]:**
*Referee comment 1*
**1. General comment**

*1. I suggest add an example figure of power spectral density in order to see in which frequency range you estimate the slope and whether the oscillations affect the slope estimation. An exact estimation of the slope is quite important to the results and conclusions. To me, the Hurst value=0.06 is really strange.*

**2. Specific comments**
**2.2 Methodology**

*β is estimated by the slope of the power-spectral density function, it must be valid in a certain frequency range. So it is better to present an example figure of a power spectral density of one station. Then it is much clearer to see how you estimate the slope, which frequency range it is and if the oscillations influence the estimation of the β. Like Figure 2 in Shao and Ditlevsen., 2015, Figure 1-3 in Pelletier and Turcotte., 1997.*

**Referee comment 2**

1. Since it is long-range correlation analysis, there should be the plots showing how to fit the scaling exponent, which range has been used. This kind of important figures are not given in the manuscript. The conclusions given in this manuscript may be not believable.

[revised manuscript text omitted]

**Commented [U13]:**
*Referee comment 1*
**2. Specific comments**
**2.5 Results and Discussion**

For BLC station, during active period, the minimum Earth magnetic field strength is 58790, but the H(BLC)=0.6466. For SIT station, the minimum Earth magnetic field strength is 55750, but the H(SIT)=0.3517. It is completely different from your result.
I suggest add an explanation about that. And the Hurst exponents of SIT and FCC stations during active period are lower than them during the normal period, which are totally different from other stations. Also an explanation for it.

Figure 7 represent the periodogram for mid latitude region during active period and normal period. The mid latitude region is situated in $30^0$ to $60^0$ latitude.

[Figure]

**Figure 7: Periodogram for mid latitude region during active period and normal**

For mid latitude region, table 4 shows the results of Hurst exponent value during geomagnetic storm (active period) and normal period.

**Commented [U14]:**
*Referee comment 1*
**1. General comment**

*1.I suggest add an example figure of power spectral density in order to see in which frequency range you estimate the slope and whether the oscillations affect the slope estimation. An exact estimation of the slope is quite important to the results and conclusions. To me, the Hurst value=0.06 is really strange.*

**2. Specific comments**
**2.2 Methodology**

$\beta$ is estimated by the slope of the power-spectral density function, it must be valid in a certain frequency range. So it is better to present an example figure of a power spectral density of one station. Then it is much clearer to see how you estimate the slope, which frequency range it is and if the oscillations influence the estimation of the $\beta$. Like Figure 2 in Shao and Ditlevsen., 2015, Figure 1-3 in Pelletier and Turcotte., 1997.

**Referee comment 2**

1. Since it is long-range correlation analysis, there should be the plots showing how to fit the scaling exponent, which range has been used. This kind of important figures are not given in the manuscript. The conclusions given in this manuscript may be not believable.

Table 4: Hurst exponent, the maximum and minimum Earth's magnetic field strength during geomagnetic storm (active period) and normal period for station in mid latitude region.

| Station | Active Period | | | Normal Period | | |
|---|---|---|---|---|---|---|
| | The Hurst exponent value | Minimum Earth magnetic field strength (nT) | Maximum Earth magnetic field strength (nT) | The Hurst exponent value | Minimum Earth magnetic field strength (nT) | Maximum Earth magnetic field strength (nT) |
| FRN | 0.6028 ± 0.0508 | 48760 | 48800 | 0.5779 ± 0.0683 | 48780 | 48810 |
| VIC | 0.6783 ± 0.0722 | 54200 | 54330 | 0.5352 ± 0.0624 | 54270 | 54280 |
| NEW | 0.6075 ± 0.0639 | 55170 | 55320 | 0.5342 ± 0.0632 | 55250 | 55260 |
| SIT | 0.3571 ± 0.0622 | 55750 | 56320 | 0.5055 ± 0.0680 | 56050 | 56070 |
| FCC | 0.3980 ± 0.0597 | 58410 | 59260 | 0.7066 ± 0.0561 | 58940 | 58980 |

5       In the mid latitude regions, Earth's magnetic field values are not as strong as the high latitude region. The minimum and maximum values for the Earth's magnetic field during the active period were in the range of 48760 nT to 58410 nT and 48800 nT to 59260 nT, respectively. During normal periods, the minimum and maximum values of the Earth's magnetic field ranged from 48780 nT to 58940 nT and 48810 nT to 58980 nT, respectively. The Hurst exponent value showed a mixture of persistent and antipersistent values. SIT and FCC stations showed antipersistent values during active period. The value were

10    0.3571 ± 0.0622 and 0.3980 ± 0.0597. During normal period, the value was persistent with 0.5055 ± 0.0680 and 0.7066 ± 0.0561. As for VIC stations, NEW and FRN showed persistent values during active period and normal period.

       It can be said that the value obtained in the mid latitude region differs from the value in the high latitude region during normal period and the active period may be due to different Earth's magnetic field values. Minimum Earth's magnetic field for mid latitude region during active period and normal period is from 48760 nT to 58410 nT and 48780 nT to 58940 nT,

15    respectively. For the maximum Earth's magnetic field values were 48800 nT to 59260 nT during active period and 48810 nT to 58980 nT during normal period. This is in contrast to the high latitude region that has a higher Earth's magnetic field.

In addition to the Earth magnetic field, latitude position can also affect Hurst exponent. It can be seen that station NEW, VIC and FRN are nearer to the SAA region, thus affecting the value of Hurst Exponent to be persistent. This is because in the SAA region, the value of Hurst exponent tends to be persistent. For SIT and FCC stations are located far from the SAA region and closer to the high latitude region where the region is more vulnerable to be antipersistent.

Figure 8 indicate the periodogram for SAA region during active period as well as normal period.

[Figure]

**Figure 8: Periodogram for SAA latitude region during active period and normal period.**

For SAA region, table 5 reveal the results of Hurst exponent value during geomagnetic storm (active period) and normal period.

**Commented [U15]:**
*Referee comment 1*
**1. General comment**

*2. There are still stations which are against your conclusion, like SIT and FCC station. As I see, the results of these two stations are totally different from your conclusion. I suggest add more explanation about that.*

**2. Specific comments**
**2.5 Results and Discussion**

For BLC station, during active period, the minimum Earth magnetic field strength is 58790, but the H(BLC)=0.6466. For SIT station, the minimum Earth magnetic field strength is 55750, but the H(SIT)=0.3517. It is completely different from your result.

I suggest add an explanation about that. And the Hurst exponents of SIT and FCC stations during active period are lower than them during the normal period, which are totally different from other stations. Also an explanation for it.

**Commented [U16]:**
*Referee comment 1*
**1. General comment**

*1.I suggest add an example figure of power spectral density in order to see in which frequency range you estimate the slope and whether the oscillations affect the slope estimation. An exact estimation of the slope is quite important to the results and conclusions. To me, the Hurst value=0.06 is really strange.*

**2. Specific comments**
**2.2 Methodology**

$\beta$ is estimated by the slope of the power-spectral density function, it must be valid in a certain frequency range. So it is better to present an example figure of a power spectral density of one station. Then it is much clearer to see how you estimate the slope, which frequency range it is and if the oscillations influence the estimation of the $\beta$. Like Figure 2 in Shao and Ditlevsen., 2015, Figure 1-3 in Pelletier and Turcotte., 1997.

**Referee comment 2**

1. Since it is long-range correlation analysis, there should be the plots showing how to fit the scaling exponent, which range has been used. This kind of important figures are not given in the manuscript. The conclusions given in this manuscript may be not believable.

[revised manuscript text omitted]

Line 5, 'This is happening in low latitude region.' You mean the mid latitude region?

**Commented [U18]:**
*Referee comment 1*
**1. General comment**

*2. There are still stations which are against your conclusion, like SIT and FCC station. As I see, the results of these two stations are totally different from your conclusion. I suggest add more explanation about that.*

**Referee comment 2**

3. The strength of the Earth's magnetic field influences the estimated spectral exponent \beta over different regions and different phases cannot explain all the results given in Tables 3-5, exceptional results for some specific stations should be explained.

**Competing Interests**

The authors declare that they have no conflict of interest.

**Acknowledgements**

The results presented in this paper rely on data collected at magnetic observatories. We thank the national institutes
that support them and INTERMAGNET for promoting high standards of magnetic observatory practice
(www.intermagnet.org).

**References**

[revised manuscript text omitted]

---

## Author Response (AR2)

**Response to referee comments**

**Manuscript title:** Characterization of the South Atlantic Anomaly

**Authors:** Khairul Afifi Nasuddin, Mardina Abdullah, Nurul Shazana Abdul Hamid

We thank the reviewers and editor for their comments on the manuscript. We have taken into consideration in preparing our revision and made minor amendment/additions of sentences to improve the revised manuscript. All lines and page numbers mentioned are referring to the revised manuscript (Track Changes-Simple Markup).

**Revised Submission**

**(1) Comments from Referees:** *Page 7, line 5, ' it can interpret as both that ' should be ' it can be interpreted…'*

**(2) Author's response:** We have change it at page 7, line 5.

**(3) Author's changes:** If the Hurst exponent is in the range of 0.5<H<1, it can be **interpreted** as both that a high value in the series will probably be followed by another high value also that the values a long time into the future will serve to be high.

**(1) Comments from Referees:** *Page 8, line 8, ' Active period can be define as a day ' should be ' be defined as…'*

**(2) Author's response:** We have change it at page 8, line 8.

**(3) Author's changes:** Active period can be **defined** as a day from 1 UT until 24 UT where the existence of the geomagnetic storm is below – 30 nT.

**(1) Comments from Referees:** *Page 9, line 3, ' The explanation on choosing 11 March 2011 compare to the other date can be explained more detail by … ' should be ' compared to the other date can be explained in more detail by…'*

**(2) Author's response:** We have change it at page 9, line 3.

**(3) Author's changes:** The explanation on choosing 11 March 2011 **compared** to other date can be explained **in** more detail by referring Fig. 4

**(1) Comments from Referees:** *Page 10, line 4, ' The periodogram is on 03 February 2011 station THL' should be ' The periodogram is obtained from station THL on 03 February 2011'.*

**(2) Author's response:** We have change it at page 10, line 7 and line 8.

**(3) Author's changes:** The periodogram **is obtained from station THL on 03 February 2011**.